# In Vivo Electroporation of Plasmid DNA: A Promising Strategy for Rapid, Inexpensive, and Flexible Delivery of Anti-Viral Monoclonal Antibodies

**DOI:** 10.3390/pharmaceutics13111882

**Published:** 2021-11-06

**Authors:** Silvere Pagant, Rachel A. Liberatore

**Affiliations:** RenBio, Inc., Long Island City, New York, NY 11101, USA; silvere.pagant@renbio.com

**Keywords:** monoclonal antibodies, plasmid DNA, electroporation, antiviral, neutralizing antibody, infectious disease

## Abstract

Since the first approval of monoclonal antibodies by the United States Food and Drug Administration (FDA) in 1986, therapeutic antibodies have become one of the predominant classes of drugs in oncology and immunology. Despite their natural function in contributing to antiviral immunity, antibodies as drugs have only more recently been thought of as tools for combating infectious diseases. Passive immunization, or the delivery of the products of an immune response, offers near-immediate protection, unlike the active immune processes triggered by traditional vaccines, which rely on the time it takes for the host’s immune system to develop an effective defense. This rapid onset of protection is particularly well suited to containing outbreaks of emerging viral diseases. Despite these positive attributes, the high cost associated with antibody manufacture and the need for a cold chain for storage and transport limit their deployment on a global scale, especially in areas with limited resources. The in vivo transfer of nucleic acid-based technologies encoding optimized therapeutic antibodies transform the body into a bioreactor for rapid and sustained production of biologics and hold great promise for circumventing the obstacles faced by the traditional delivery of antibodies. In this review, we provide an overview of the different antibody delivery strategies that are currently being developed, with particular emphasis on in vivo transfection of naked plasmid DNA facilitated by electroporation.

## 1. Monoclonal Antibodies as Antivirals

Monoclonal antibodies (mAbs) are now recognized as a major tool in the clinical arsenal for therapies against a wide variety of ailments, including cancer, inflammatory diseases, and cardiovascular diseases. Antibody therapies are also becoming increasingly important for the treatment and prevention of infectious diseases. To date, mAbs against SARS-CoV-2 are the only treatments clinically proven to prevent hospitalization and death for individuals infected by or exposed to the virus. The main attribute that makes mAbs both powerful prophylactics and therapeutics is their ability to neutralize virus entry by binding, with high specificity and avidity, to the surface proteins of the virus that engage receptors on the surface of host cells. Moreover, mAbs can also prevent viral spread through the recruitment, mediated by their Fc (fragment crystallizable) domain, of additional immune cells to infected cells, thereby triggering their removal via a variety of mechanisms such as antibody-dependent cell-mediated cytotoxicity (ADCC), antibody-dependent cellular phagocytosis (ADCP), and complement-dependent cytotoxicity (CDC) [1,2]. In the context of an outbreak of a novel virus, passive administration of mAbs represents an ideal strategy for combating viral diseases for which a vaccine has not yet been developed. It can also be a powerful alternative to stop rapid outbreaks by bypassing the time required to develop effective immunity through traditional vaccination, and importantly, it can also be used to protect people who do not develop a robust vaccine response due to an immune system vulnerability.

Given the incredible potential of mAbs as drugs, the last decade has witnessed an explosion in the efforts to characterize mAbs against a variety of pathogenic viruses, including different strains of influenza virus [3,4], human immunodeficiency virus 1 (HIV-1), cytomegalovirus (CMV), hepatitis C virus (HCV), Ebola virus, Marburg virus [5,6], dengue virus [7,8], Hendra and Nipah viruses [9,10,11], yellow fever virus [12,13], and West Nile virus [14]. Remarkably, just in the past year and a half, more than 60 antibodies binding the spike protein of SARS-CoV-2 that have been isolated from COVID-19 patients [1,15,16,17,18,19,20,21,22,23,24,25,26,27,28,29,30,31,32,33] have demonstrated high potency in neutralizing the virus ex vivo and in protecting against virus infection and virus spread post-infection in preclinical models such as mice, hamsters, and nonhuman primates (NHPs) (reviewed in [34]). Some of these mAbs are currently being tested in the clinic, and four combinations have received emergency use authorization (EUA) from the FDA for the treatment of individuals testing positive for the virus.

The rapid rise in the development of antiviral antibodies stems from the tremendous technological advancements in their identification and isolation (notably single B cell sorting coupled with antibody sequencing) as well as in their manufacturing. These advancements have enabled the discovery and production of very potent molecules in record time. As an example, a cocktail of two mAbs developed by Regeneron received a EUA by the FDA on 21 November 2020 for the treatment of COVID-19, only 8 months after the disease had been declared a pandemic by the World Health Organization (WHO).

## 2. Limits and Nucleic Acid-Based Alternatives

Despite these many advances in mAb technology, the global deployment of mAbs as biologic drugs, which would be needed to fight pandemics such as COVID-19, is still limited. Large-scale manufacture of antibodies remains a highly technical, somewhat lengthy, and expensive process due to their complex biochemical and biophysical properties. Indeed, this complexity can preclude the clinical development of certain molecules with great potency and efficacy but poor developability characteristics. Delivery, distribution, and access are also important challenges to consider for the widespread use of mAbs as antivirals.

Historically, mAbs have been primarily administrated by intravenous (IV) infusions, which are generally conducted in special clinical settings and come with the associated cost and training of specialized personnel. Such infusions typically take several hours in total, which might hamper their use in geographic areas or communities with limited resources. More amenable routes of administration, such as intramuscular (IM) and subcutaneous (SC) injections or intranasal (IN) delivery for nebulized mAbs, are currently under investigation for antiviral mAbs. Indeed, the SC route has proven to be increasingly effective for a number of mAbs currently in use for oncology. Importantly, the use of alternative routes of administration needs to be evaluated almost on a mAb-by-mAb basis, as the usefulness of each mode of injection depends on the effective dose required and the biodistribution necessary for each mAb to achieve clinical benefit. Finally, due to the rather short half-life of mAbs once injected, pre-exposure prophylaxis or sustained post-exposure therapy, if necessary, would require repeated dosing, which multiplies the associated challenges mentioned above with each cycle of injection.

For these reasons, there is growing interest in developing alternative strategies for easier delivery and improved durability of mAbs, particularly at reduced costs, in order to provide wider access to these powerful biologic drugs. Among the different approaches studied, nucleic acid-based technologies seem to be the most advanced and the most promising (Figure 1). Such technologies are based on the in vivo transfer of genes encoding antibodies into host cells, which, in turn, serve as a bioreactor by synthesizing and secreting the therapeutic protein from the blueprint provided by the delivered DNA or mRNA. Of interest, both viral and non-viral gene transfers have been shown to provide therapeutic levels of mAb for an extended period of time as compared to passive immunization with the purified protein.

### 2.1. Adeno-Associated Virus Delivery Technology

Adeno-associated virus (AAV) vectors are currently the most frequently used delivery system for viral-mediated gene transfer in the expanding field of gene therapy, though AAV has also been evaluated for the expression of antiviral mAbs. The popularity of AAV vectors for multiple gene delivery purposes is largely due to their capacity to transduce both dividing and post-mitotic cells and from the absence of integration of their vectors into the host genome, limiting putative negative off-target outcomes.

Numerous preclinical studies have shown that a single IM injection of an AAV vector encoding a neutralizing antiviral mAb results in persistent circulating levels of the antibody at concentrations sufficient to protect against infection. This approach was first pioneered in the HIV-1 field, where it was shown to be efficient in preventing disease in humanized mice caused by systemic or mucosal HIV-1 infection. AAV-mediated delivery of neutralizing antibodies (NAbs) also conferred sterilizing protection against systemic simian immunodeficiency virus (SIV) infection when applied to NHP models [35,36,37]. Remarkably, in an extended longitudinal study of a unique monkey that did not develop a typical anti-drug antibody (ADA) response against an AAV-delivered human NAb, the animal displayed robust protection from repeated challenges with increasing doses of a highly pathogenic strain of SIV, 6 years after the initial AAV administration [38,39]. AAV-mediated delivery of mAbs has now been tested in mice and NHP models for many other viral diseases such as influenza [40,41], dengue virus [42], Ebola virus [43], and respiratory syncytial virus (RSV) [44], demonstrating the theoretical universality of this strategy for persistent protection against various types of viruses.

Widespread translation into the clinic might have to wait for further optimization of AAV technology, however, as very low mAb expression was reported for one of the ongoing clinical trials aimed at the prevention of HIV-1 infection (NCT01937455) [45]. Interestingly, the use of another HIV-1 NAb and different AAV serotype in a more recent clinical trial (NCT03374202) seemed to generate more promising preliminary results, suggesting that the antibody selection, AAV serotype, and expression strategy, among other things, need to be carefully evaluated to achieve clinical relevance. These considerations are subjects of intense research in the AAV field in general, and the final outcome of these two clinical trials, and of the myriad others employing AAV-mediated gene transfer for other disease areas, will certainly bring valuable information for future studies.

In addition to expression-related concerns, the occurrence of deleterious adverse events (including death) during recent clinical trials evaluating AAV-mediated gene replacement for rare diseases (NCT03199469) is also prompting the AAV field and the regulatory agencies to seriously reassess the safety of such therapies. This will no doubt slow down their advancement until these important issues are better understood and resolved. Moreover, the generally observed immune response to AAV vectors might also limit their use for antibody delivery as it hampers re-dosing scenarios that might be required in cases where therapeutic levels cannot be achieved with a single injection (either due to low expression in the first place or due to waning expression over time) or, in the case of an evolving viral threat, when delivery of a new mAb might be required to counter the emergence of a new variant resistant to initial treatment. Lastly, the manufacturing of clinical-grade AAV vectors remains a lengthy and costly process, which restricts their use on a large scale, including for pandemic prevention strategies.

### 2.2. mRNA-Based Delivery

Non-viral delivery systems for nucleic acids encoding therapeutic proteins are the subject of growing interest as alternatives to viral vectors such as AAV. Because such systems do not elicit anti-vector antibodies, they are typically more amenable to repeat dosing and avoid the efficacy and safety concerns that often accompany such immune responses.

The recent technical and medical success of messenger RNA (mRNA) vaccines against COVID-19 is a testament to the tremendous progress that has been made in the transition of lipid nanoparticles (LNP)-formulated mRNA to clinical settings. In contrast to vaccines, however, where the mRNA molecules encode a viral antigen, similar technology can also be used for passive immunization against infectious agents in strategies where the mRNA encodes a previously identified NAb. Preclinical studies have demonstrated the potential and feasibility of such methods for protection against multiple viruses, both from a treatment and prevention standpoint [46,47,48]. A single injection of mRNA-encoding anti-rabies mAbs, packaged in LNPs, was able to provide both prophylactic and therapeutic protection against viral challenge in mice [46]. In the same study, a neutralizing antibody against influenza B could also be detected at comparable levels in the circulating blood of a mouse cohort treated with the corresponding mRNA. Protection against SF162 and JR-CSF HIV-1 challenges was also observed in humanized mice receiving the mRNA coding for the HIV-1 NAb VRC01, yielding similar efficacy as to what was observed following treatment with the corresponding recombinant protein [47]. Finally, IV injection of the mRNA coding for an anti-CHIKV mAb provided dose-dependent protection in mice challenged with the virus, and the same LNP-mRNA formulation was able to produce therapeutic levels of the antibody in cynomolgus macaques [48].

These results, along with the efficacy of COVID-19 mRNA vaccines, make LNP-mRNA technologies very attractive in the fight against viral disease. However, this strategy suffers from several limitations to its broad use as a platform for expressing antiviral mAbs. Due to its chemical structure and susceptibility to degradation, mRNA has a short half-life in vivo. This is illustrated by the aforementioned preclinical studies, which showed that high levels of NAbs in mice were achieved for only a short period of time (1–2 weeks on average) after a single injection of mRNA and that sustained expression required weekly dosing. In addition, foreign mRNA molecules are rapidly detected by a suite of sensors (such as toll-like receptors 3, 7, and 8) following cell entry, and this detection triggers swift and strong inflammatory responses. In the context of active vaccination, these responses are welcome and can serve as an adjuvant, bolstering vaccine-induced immunity. However, these same responses could be major drawbacks for other clinical applications, such as antibody therapy, and have the potential to contribute to serious adverse events, mostly in cases when high doses are necessary to achieve therapeutic protein levels.

The clinical trial (NCT03829384) of an mRNA-based therapy developed by Moderna for the expression of an anti-CHIKV mAb is underway, and represents the first such study. The preliminary results of this phase I trial, evaluating the safety, tolerability, pharmacokinetics (PK), and pharmacodynamics (PD) of a dose escalation (from 0.1 to 1 mg/kg), are reflective of the limitations of this technology. Recipients of the therapy expressed high levels of the mAb, even for the low dose regimen, with a peak at 24 h post-injection followed by a stable decline. Detectable levels were however present at least 84 days after treatment in people who had received middle range dose (0.3 mg/kg). Three out of four participants at the high (0.6 mg/kg) dose presented infusion-related adverse events, despite antihistamine premedication. This trial is overall encouraging for mRNA-based antibody therapies, and the very recent partnership between Moderna and Abcellera, a company focused on antibody discovery, suggests that there is continued interest in this approach. However, this initial clinical study also highlights the improvements in efficacy and safety needed for more widespread use. Additionally, LNP-formulated mRNA for this type of application suffers from some of the same logistical drawbacks as standard mAb protein administration; it is delivered through IV infusion and needs cold-chain infrastructure for storage and transport, again limiting deployment and accessibility to low-resource settings.

### 2.3. Electroporation of Plasmid DNA

Compared to protein, AAV, and mRNA-based therapies, synthetic plasmid DNA (pDNA) offers several advantages, such as flexible sizes, low immunogenicity, low production and storage costs, and distribution that does not require a cold-chain. Similar to mRNA, pDNA has been evaluated for use in vaccines in order to express a given antigen [49,50,51,52,53]. Comprehensive studies on pDNA vaccine platforms for different viruses (HIV-1, SARS-CoV, West Nile virus, and Ebola virus) conducted in mice or rabbits have shown that the injected pDNA remains at the site of injection for a mean period of two months, followed by eventual clearance in the majority of animals. These features were observed regardless of the coding and non-coding sequences present in the plasmid, and no genomic insertion was observed, demonstrating the safety and suitability for investigational human use of pDNA for a variety of infectious disease prevention indications [54]. Importantly, pDNA is stable at ambient temperature for extended periods of time and can withstand significant temperature fluctuation without damage in most standard buffer formulations. Additionally, because pDNA does not induce anti-vector immune responses, a homologous synthetic DNA backbone encoding the same or a different gene can be repeatedly readministered. Due to these benefits, plasmid DNA represents 14.7% (*n* = 482) of vectors used for gene transfer in gene therapy clinical trials world-wide (GTCT (FMS19) (fmphost.com) (accessed on 1 October 2021).

Direct injection of pDNA leads to some expression of the encoded transgene, but the levels are significantly lower than what is observed with viral vectors and generally considered insufficient for therapeutic relevance [55,56,57]. Electroporation (EP) is becoming the preferred method to improve cellular entry for in vivo gene transfer of pDNA (Figure 2). In the laboratory, EP has been used for decades to trigger the entry of DNA into bacteria, yeast, and mammalian cells. More recently, EP has been developed to increase in vivo delivery of pDNA to tissues, primarily skin and skeletal muscle, following a direct SC or IM injection. This technique entails the application of electrical fields to permeabilize the cell membrane to macromolecules by generating transient pores, and further facilitates DNA entry into cells through electrophoresis [58]. EP can improve DNA entry in muscle fibers up to 100–1000 fold in various muscles in the mouse, rat, rabbit, and monkey [56,57], including both the number of cells transfected and the level of DNA uptake.

As described for AAV strategies, once present in living cells of the patient, pDNA can be used as a template to produce the protein(s) encoded within it. This method is currently used to develop DNA vaccines via EP-mediated delivery of antigen-encoding pDNA, which will in turn trigger an immune response. Similar methods are also being investigated for the production of secreted proteins, including antibodies, and have begun to demonstrate attractive preclinical results for the prevention and/or treatment of viral diseases.

The key optimizations of EP for use as pDNA delivery method are centered around defining the optimum electrical conditions for maximal plasmid entry with minimal muscle damage [55,56,57,59,60], as exposure to some types of electrical pulses can also lead to cell death [61]. Several studies have shown that multiple variables of the electrical field applied need to be considered, including voltage or amperage, pulse duration, pulse number, and lag time between pulses [60,62,63]. Early studies have also shown that identical plasmid doses are more efficiently transfected into muscle cells if delivered in larger volumes [64,65], outlining the importance of plasmid dispersion within the electroporated area [66].

In addition to the optimization of technical determinants leading to robust pDNA entry, engineering and formulation of plasmid vectors have been keys to improving the efficacy of this platform. Not surprisingly, mAb expression can vary substantially depending on regulatory sequences present in the pDNA backbone. Human CMV promoter and enhancer sequences [67], as well as the chicken beta-actin promoter [68], are the elements of choice in most studies since they are known to provide strong and constitutive transgene transcription in vivo. Sequences in polyadenylation signals, the 5′ and 3′ untranslated regions (UTRs), and other response elements characterized to modulate transcription, mRNA stability, or translation, are also means to provide benefits. The nature of the plasmid backbone itself is clearly significant, as shown by Andrews et al., where the same mAb gene expressed from the pVAX plasmid, traditionally used for DNA vaccines, yielded lower mAb titers than when expressed from a gWiz plasmid [69]. As the pDNA-mAb platforms advance, other variations of the plasmid backbone, including closed linear DNA, minicircles, and nanoplasmids [70], are being evaluated for their ability to expand the potency and safety of this technology.

Progress can also be achieved via alterations in the specific formulation of the pDNA to be electroporated. Pre- or co-treatment with hyaluronidase, a degrative enzyme of one of the major components of the extracellular matrix (ECM), greatly improves SC and IM administration of recombinant antibodies and has been shown to enhance EP-mediated pDNA delivery in many preclinical studies in mice [60] and larger animals [71,72]. Indeed, hyaluronidase is currently a component of the pDNA formulation in the only clinical trial investigating EP for the delivery of mAbs in humans (NCT03831503). The utilization of other enzymes, such as chondroitinase ABC, which process other polymers present in ECM, has also been shown to be associated with higher levels of mAb production [49]. Importantly, this not only provides an alternative to hyaluronidase but also establishes the distribution of pDNA within the muscle tissue as an important component of these technologies. In further support of this idea, some groups have found a benefit from the physical distribution of the pDNA by delivering multiple injections [73].

A common roadblock to the development of standard protein antibody therapies is the presence of one or more certain unfavorable bio-physical features, such as amino acid oxidation, deamination, or isomerization, that influence stability and aggregation during manufacturing as well as stability of the final product. These obstacles can have the unfortunate effect of preventing the clinical development of antibodies topping the list in terms of potency and breadth. To this point, one of the attractive promises of pDNA-based mAb delivery is that the in vivo production of the mAb sidesteps the need for in vitro manufacturing and storage. The skeletal muscle cells of the individual receiving the treatment are able to produce and secrete the therapeutic antibodies regardless of their in vitro stability characteristics. This can allow for high in vivo expression of molecules that prove to be difficult to produce by conventional approaches. This hypothesis was elegantly tested in a recent study, where the authors compared the pDNA-driven expression in mice of eight mAbs targeting the glycoprotein of the Ebola Virus, with their predicted DI (developability index) in silico [74]. Interestingly, mAbs with a poor predicted DI could be expressed in vivo at high levels, confirming the utility of EP-administered pDNA for rapid in vivo screening of mAbs and, importantly, highlighting the potential for clinical transfer of potent mAbs that would otherwise be difficult to develop by conventional means.

## 3. Electroporation of Plasmid DNA for the Delivery of Antiviral mAbs

Several preclinical studies have established proof-of-principle functional data for antiviral mAb delivery using EP-mediated administration of NAbs targeting dengue virus [75], different strains of influenza viruses [69,76], Ebola virus [74], Zika virus [72,77], CHIKV [78], RSV [79], Hepatitis B virus [80] and HIV-1 [79,81].

These studies first demonstrated that high levels of antibody in circulating blood can be obtained for weeks to months after single dosing of pDNA. In addition, mAbs present in the sera of animals present the same binding affinity for their cognate antigen as compared to the corresponding recombinant mAb [72,77]. Patel et al. also demonstrated, using alanine scanning mutagenesis of the target protein, that in vivo-produced and recombinant mAb detect the exact same epitope at the amino acid residue level. Accordingly, the degree of ex vivo neutralization of the targeted virus was shown to be equivalent between the pDNA-derived mAb and the corresponding purified protein [72,78,80,81]. When using the appropriate mouse model for each specific pathogen, such pDNA-based expression of NAbs was able to provide robust protection against lethal viral challenges as early as 2–5 days after to EP [72,75,76,77,78] and up to 23–31 days [69,78]. In addition, this technology proved to be effective against both systemic and mucosal infection with CHIKV via subcutaneous or nasal challenges, respectively [78]. Application of the same strategy to other animal models has been equally promising; in vivo EP-mediated delivery of a NAb in cotton rats protected them from RSV infection [79] and significantly reduced the viral load of NHPs challenged with Zika virus [72]. In addition to providing important proof-of-concept data for the utilization of this approach in the fight against various viral infections, these studies also provide interesting evidence about its potential advantages over other antibody administrations.

### Neutralizing Antibody Delivery via Electroporation vs. Vaccination

As previously mentioned, active immunization by vaccination, although the ideal tool to protect large populations in the long term, is not suitable for rapid responses to new infectious agents since the immune response triggered by vaccination, and needed for protection, occurs at a minimum a few weeks post immunization. Furthermore, maximum levels of defense are often only achieved through one or more additional vaccine boosts. Aside from the timing of the protection afforded by vaccines, they also rely on a robust immune response from the individual to mount the necessary titer of neutralizing antibodies and other elements of adaptative immunity, which might exclude certain populations with weak or suppressed immune systems. For example, a longitudinal study examining the efficacy of COVID-19 mRNA vaccines in solid organ transplant recipients under immunosuppression regimens (such as tacrolimus, corticosteroids, mycophenolate, azathioprine, sirolimus, or everolimus) revealed that anti-spike antibodies were detected in only 17% of the participants after one dose and 35% responded only after the second dose, with only 8% responding to both doses [82,83]. COVID-19 vaccines were also inefficient at triggering appropriate immune responses in other patient cohorts undergoing immunosuppression treatments, such as patients with rheumatic and musculoskeletal diseases [84], heart and lung transplant recipients [85], and patients with multiple sclerosis [86]. Indeed, antibodies are currently the best therapeutic option for providing these individuals with protection against viral infections.

The effective use of EP-mediated pDNA administration to deliver mAbs as therapeutics post-exposure will depend on both the kinetics of the disease course as well as the potency of the mAb. Peak serum mAb levels are achieved typically within 2–3 weeks following administration, which is more compatible with treating viral infections with a longer clinical viral replication phase, if peak serum mAb levels are required for efficacy. If an antibody is sufficiently potent, therapeutic mAb levels could be reached much more quickly and would be relevant for treating acute viral infections. Similarly, effective concentrations of a potent mAb could be reached in a week or less for prophylactic purposes.

A pair of studies in mice evaluated the protection offered by EP-mediated pDNA delivery of a mAb as compared to a DNA vaccine for therapies directed against either CHIKV [78] or Zika virus [77]. The first study compared the efficacy of a plasmid coding for either CHIKV ENV (the vaccine) or a NAb that binds to the CHIKV envelope protein, both delivered by EP. All the mice subjected to the passive immunization with pDNA encoding the NAb survived when challenged with virus 2 days after electroporation, whereas 100% of the group that had received the DNA vaccination regimen died. These data illustrate the superiority of DNA-based delivery of Nabs for health care strategies designed to confer rapid protection and halt an outbreak of a highly transmissible virus such as SARS-CoV-2. Conversely, 100% of the vaccinated mice survived, compared to 80% for the cohort dosed with the plasmid encoding the mAb, upon a later challenge at 35 days post EP. It is worth noting that the vaccinated cohort received three rounds of DNA immunization, which might explain their enhanced protection, whereas the protection conferred by the antibody therapy resulted from a single injection regimen. It is also important to mention that the levels of the encoded human antibody produced might be underrepresented, both in terms of titer and duration, due to the presumed reduction of its half-life from its expression in mice. The application of this technology for the expression of a human antibody in humans would be expected to result in even more potent and lasting protection. Finally, co-delivery of both plasmids prior to electroporation protected mice from both day 2 and day 38 challenges. Similarly, the co-EP of plasmids coding for a Nab, on the one hand, and an antigen, on the other hand, protected mice from morbidity and mortality following early (day 2) or late (week 3) inoculations with Zika virus. These findings support the idea that delivery of pDNA by EP presents an opportunity to design highly effective, low-cost, and sustainable strategies for the fight against viral pandemics, which would otherwise require the deployment of multiple technology platforms.

## 4. Antibody Cocktails

Combinations, or cocktails, of two or more mAbs that target multiple epitopes present on the surface protein(s) of a given virus are quickly becoming the preferred strategy to tackle infectious diseases. This approach, by simulating the polyclonal immune response triggered by natural infection or the polyclonal antibody therapy that can be provided by convalescent plasma, offers several safeguards for broad and sustained protection against viruses. Such a strategy also provides a higher barrier to the emergence of escape mutations in the target virus, unlike monotherapies, which can exert a selective pressure that can result in viral variants of the specific epitope targeted, as has been reported with HIV-1 [87] and Ebola [88].

Indeed, antibody cocktails support a wider range of defense from different circulating variants and strains at any given time, making this strategy particularly important when faced with vast amounts of virus, as is the case during a pandemic. Supporting this notion, mixtures of two to four antibodies binding different neutralizing epitopes on the HIV-1 envelope glycoprotein were able to neutralize 100% of viruses from a panel of 125 strains under conditions in which the individual antibodies only neutralized 25–66% [89]. This breadth is particularly critical for continued protection against viruses such as SARS-CoV-2, influenza, and HIV-1, for which new variants that become resistant to a specific mAb often emerge. As an example, the clinical use of a mAb directed against SARS-CoV-2 developed by Eli Lilly (bamlanivimab) was halted due to the emergence of the SARS-CoV-2 variants alpha and beta, which carry amino acid substitutions in their spike protein that are located in the epitope recognized by this antibody. Likewise, these changes also negatively impact the function of one of the mAbs in the cocktail developed by Regeneron, but the use of this cocktail for the treatment of COVID-19 was continued because the activity of the other mAb in the cocktail was unaffected, highlighting the advantage of a multi-pronged approach.

Although cocktail-based mAb therapeutics have the potential to be the most broadly effective strategy for combating many viral diseases, when they are applied as a traditional protein therapy, the cost of production is multiplied by the number of antibodies present in the final drug product. Due to the relatively easy and inexpensive manufacture of pDNA, however, EP-based pDNA delivery of multiple mAbs presents a more attainable alternative. In addition, unlike what is observed for AAV vectors, the absence of an immune response against the plasmid backbones allows the administration of several plasmids encoding mAbs targeting different epitopes without the risk of cross-inhibition. This technology was first tested in mice using anti-dengue virus mAbs [75] and showed that EP with IM administration of 2 plasmids encoding antibodies directed against different viral epitopes increased the titer of total mAbs in the circulating blood and increased the breadth of coverage to all four dengue serotypes tested. Similarly, subsequent studies in mice have confirmed the feasibility of DNA-EP in delivering multiple plasmids to express antibody cocktails to improve protection against other types of viruses, namely different strains of influenza [69,76].

## 5. Alternative Antibody Formats

The field of antibody engineering has witnessed significant advancements in recent years, with a broad range of novel protein domains and formats being explored. In some cases, pDNA-based delivery of such novel formats has the potential to greatly facilitate their successful translation to the clinic.

First, it is extremely straightforward to alter a pDNA sequence to introduce protein changes that confer therapeutic advantages, including modifications that extend a mAb’s half-life in vivo. Moreover, as previously mentioned, in addition to their neutralizing function mAbs can also prevent disease due to viral infection by recruiting immune cells via their Fc domain, resulting in the elimination of infected cells. However, the same Fc functions can, in certain cases, also result in antibody-dependent enhancement (ADE) of infection by facilitating virus entry. Known amino acid substitutions in the conserved Fc domain of antibodies have been characterized for their ability to enhance or prevent such functions. These changes can easily be incorporated into pDNA and then quickly tested by EP-mediated delivery in preclinical disease models to assess whether they can provide a benefit for treatment or, conversely, exacerbate the disease.

An active area of research toward the next generation of therapeutic mAbs is the clinical development of antibody fragments that retain full antigen-binding capacity. The smallest of these is the Fv (fragment variable), which comprises all the antigen-binding regions of a complete antibody molecule. This domain can be further manipulated as a modular unit to create a suite of synthetic antibody formats such as scFv (single chain Fv), which connects Fv fragments to each other via peptide linkers, or Fab (fragment antigen-binding), which includes the variable-adjacent constant region. The use of nanobodies as building blocks can further reduce the size of these types of molecules since at only 15 kDa, they comprise the entire single variable domain of the heavy chain-only antibodies found in camelids. Their small size has the potential to offer several advantages that can significantly improve antibody therapy. For example, they possess a very different PK profile as compared to full-size mAbs, and as a result, may be more suitable for the treatment of diseases for which rapid effects are beneficial. For example, expression of the Fab of an anti-CHIKV antibody in mice preceded that of the corresponding complete antibody by a few days, though both forms were also present at longer time points [78].

The reduced size of these molecules can also support additional activities, such as binding to otherwise inaccessible epitopes and improved biodistribution due to better tissue penetration. However, their smaller size also results in a much shorter half-life in vivo due to renal excretion, which significantly limits their clinical utilization. The delivery of such molecules via pDNA offers a unique solution to this drawback since it allows for the molecule to be constantly synthesized and secreted, thereby counteracting the enhanced clearance. Using this strategy, Schultheis et al. achieved high serum levels following EP of an scFv-Fc targeting RSV for at least 90 or 40 days in mice or cotton rats, respectively [79]. In addition, this molecule was also detected in lung tissue, the physiologically relevant site associated with RSV-mediated pathology. Accordingly, this resulted in an effective reduction in viral load from the lung tissues following a nasal challenge with the RSV and, importantly, did so more efficiently than the purified protein with a serum titer 10 times higher.

The understanding of the advantages that EP-mediated pDNA delivery can provide for new antibody-based formats is still in its infancy, notably for antiviral applications. There are, however, a few proof-of-principle studies in oncology that show great promise for further development. EP delivery of pDNA coding for nanobodies against DR5, a receptor for tumor necrosis factor-related apoptosis-inducing ligand (TRAIL), resulted in more persistent serum levels of the anti-DR5, and consequently a more potent and durable anti-tumor activity in a colon cancer model than was observed with infusion of the purified protein [90]. Interestingly, analogous delivery of the same nanobodies fused with serum albumin improved the serum titer but was less effective in vivo in terms of tumor disposal. Taken together, these data suggest that the superior tumor penetration of small nanobodies offers a clear therapeutic benefit, even at suboptimal concentrations. Importantly, these concentrations can be easily achieved and durably maintained with EP technology [90].

Bispecific antibodies (bsAbs) are artificial antibody-like molecules engineered to bind to two different epitopes on the same target protein or unique epitopes on two different targets. To date, they have been primarily developed for cancer immunotherapy to simultaneously bind specific epitopes present on the surface of tumor cells and epitopes on cytotoxic T-cells, thereby bringing these two targets together to promote the removal of the cancer cells [91]. The same basic strategy could be extrapolated to viral immunotherapies for clearance of infected cells by the immune system. The bispecific format has also been investigated for the prevention of viral infection as an alternative to the development of antibody cocktails. The underlying principle in this case is the same, and that is to simultaneously target multiple epitopes on the surface of the virus, thereby improving avidity and breadth [92]. As an example, by combining parts of mAbs targeting discrete epitopes on the spike protein of SARS-CoV-2, investigators were able to create a suite of bsAbs (1) with higher in vitro neutralization potency (up to 100-fold) than a cocktail of their parent mAbs, (2) that bind the surface of the spike protein by a novel mechanism, or (3) that maintain activity against an emergent variant of concern [93]. The production and manufacturing of bsAbs, however, is currently very complex since it requires the expression of the heavy chain and light chain of both parental mAbs, and is dependent on the correct association of these four protein domains. Despite recent progress in bsAb design and manufacturing that has improved the efficiency of the formation of the desired molecule, these biosynthesis hurdles make their translation to in vivo settings challenging. For these reasons, and following the trend of small synthetic forms of mAbs, researchers are now considering the clinical development of BiTEs (Bi-specific T-cell Engagers), ~55 kDa fusion proteins composed of two scFvs derived from different antibodies. Similar to the bsAb versions, one scFv is designed to bind to T cells via the CD3 receptor, and the other to a tumor target or infected cell for immunotherapies against cancer or viral infections, respectively [94,95]. Recent data support pairing such technology with an EP/pDNA platform; delivery of a plasmid encoding a BiTE targeting both CD3 and the breast cancer marker HER2 resulted in the presence of BiTE levels as high as 50 μg/mL for more than 9 months in mice [96]. New antibody formats are attracting attention and expanding on the already impressive clinical record of mAb therapies. By overcoming their rapid clearance when injected as protein, one of the biggest limitations for their development in the clinic, delivery of these molecules via EP of pDNA offers further advantages for their future growth as therapeutics.

## 6. Future Directions for Electroporation-Mediated DNA Delivery

As EP technology for antibody delivery demonstrates promising preclinical results, the focus turns to its translatability to humans. Encouragingly, similar technology has been clinically evaluated for DNA vaccine approaches and has shown favorable safety outcomes [52,97]. The main questions regarding its specific use as a clinical antibody delivery platform are whether the technical improvements established in preclinical models will result in the same benefit when applied to humans in order to achieve therapeutic levels of mAb. For example, electrical parameters shown to provide efficient transfection of pDNA in mice and rats were not successful when applied to rabbits [63]. The reasons for these observations remain unknown, but these data highlight the fact that EP procedures are not always easily transposable from one species to another and that the parameters currently used in animal models (or in humans for the very different mechanistic goals of DNA vaccination) may require further use-specific optimizations.

A major obstacle to predicting how the efficiency of this technology, and the resulting PK profile of the antibody produced, will translate from mice or NHPs to humans comes from the difference in body size and volume of circulating blood between species. Pending the outcome of the phase I clinical trial currently ongoing for the evaluation of EP-mediated delivery of pDNA encoding an anti-dengue virus mAb (NCT03831503), the use of animals such as sheep and pigs, which can reach human-like weights, are the best available models. A limited number of studies have attempted to use these species to help inform the pDNA dosage (including both the amount and volume of pDNA injected and the number of injections) and EP device conditions necessary to achieve sufficient mAb levels in larger species.

The evaluation of EP-mediated delivery of various pDNAs in sheep [71] constitutes to date the most comprehensive assessment of these issues. In these studies, the authors were able to express a fully ovine mAb with peak serum levels of 3.5 µg/mL at 4–6 weeks, which slowly declined, but remained above 1 ug/mL for 5 months and were detectable for 11 months after treatment. Since the approximate weights of the treated animals were 40 kg and 70 kg at the start and the end of the study, respectively, these results indicate that this delivery platform can result in therapeutic levels of mAbs in animals of near-human size and are certainly encouraging for future clinical applications. It should be noted, however, that these high titers were obtained from 12 separate injections and followed pretreatment with hyaluronidase, which provided a 10-fold improvement in serum mAb levels. In addition, the maximum titers were only observed under immunosuppression, due to the appearance of ADAs against the antibody, despite the fact that it was species-matched. Indeed, variable regions of mAbs derived from the same species as the recipient can still be recognized as ‘foreign’ by the immune system, a phenomenon that has been shown to alter the efficacy and safety of some clinically approved recombinant mAbs [98]. With regard to pDNA-based delivery strategies, the use of promoters active only in skeletal muscle cells to avoid gene expression in APCs has been shown to abrogate the immune responses against the protein encoded by a transgene and could be an interesting path to limit ADA responses to mAbs [99]. Furthermore, in the sheep study, the ADA response against another sheep mAb was only detected in the context of experimental conditions that led to maximum expression (high pDNA dose with hyaluronidase), indicating that the successful development of this technology will likely require a better understanding of how certain mAb sequences, the concentration of pDNA injected, and the resulting mAb plasma titers influence not only the level of expression but also the associated immune responses.

Large animal models also afford the possibility of directly investigating different human-scale EP devices. The current devices (TriGrid™ Delivery System from Ichor Medical Systems Inc., San Diego, U.S.A. and CELLECTRA^®^ 5PSP from Inovio Pharmaceuticals, Inc., San Diego, U.S.A.) utilized in the clinic for intramuscular EP each consist of an array of electrodes that surround a single central injection point, so that the electric field created covers the area of tissue containing the injected pDNA. This device design has been shown to be effective for the application of DNA vaccines but may still be suboptimal for mAb production, and various device optimizations (such as the number and arrangement of the electrodes, the volume injected, or the number of injections) should be considered [100].

## 7. Concluding Remarks

As the COVID-19 pandemic response has revealed, mAbs have a key role to play in the prevention and treatment of viral diseases, but current delivery methods limit their access. The studies presented in this review highlight the unique promise that EP-mediated delivery of mAb-encoding pDNA offers to widen access and distribution of this important class of antivirals. In addition, this technology has several potential advantages for the clinical development of next-generation antibody formats that are designed to further increase the efficacy and impact of mAb-like molecules. While still a young field, with only the first clinical study underway, this platform has enormous potential to revolutionize the prevention and treatment of infectious diseases in the years to come.

## Figures and Tables

**Figure 1 pharmaceutics-13-01882-f001:**
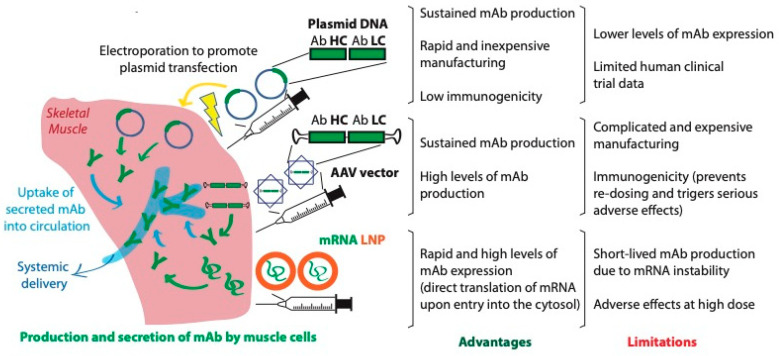
Comparison of various approaches to nucleic acid-based delivery for in vivo production of mAbs, including viral (AAV) and non-viral (electroporation of pDNA and lipid nanoparticle formulated mRNA).

**Figure 2 pharmaceutics-13-01882-f002:**
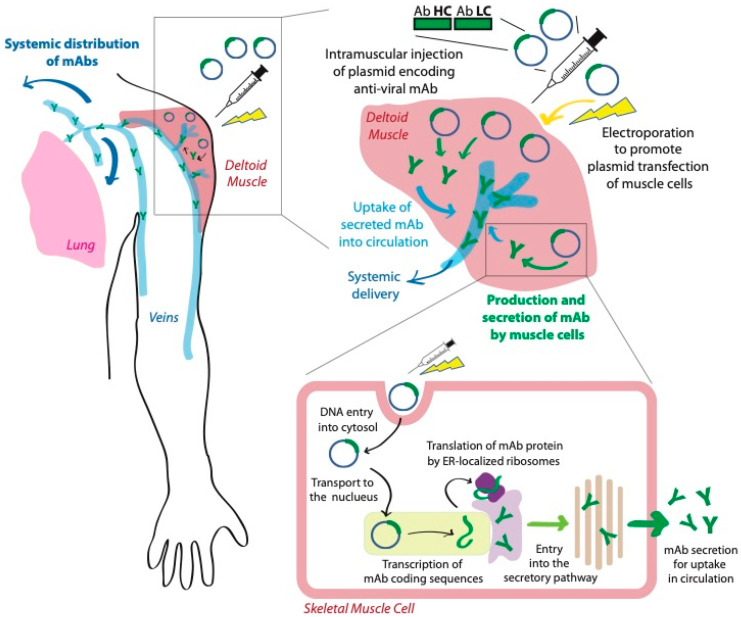
Delivery of monoclonal antibodies using electroporation-mediated administration of DNA. Intramuscular injection of plasmid DNA encoding a monoclonal antibody is followed by electroporation to promote uptake by muscle cells. Transfected muscle cells then use this genetic information to produce and secrete the encoded antibodies, which are then taken up into systemic circulation.

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
