# Peer review of "In Vivo Electroporation of Plasmid DNA: A Promising Strategy for Rapid, Inexpensive, and Flexible Delivery of Anti-Viral Monoclonal Antibodies"

_pharmaceutics, 2021, doi:10.3390/pharmaceutics13111882_

Round 1

Reviewer 1 Report

Review of the manuscript entitled “In vivo electroporation of plasmid DNA: a promising strategy 1 for rapid, inexpensive, and flexible delivery of anti-viral mono-2 clonal antibodies” (#1425077).

          This review overviews the currently developed antibody delivery strategies. This article highlights a little-used technique of preventing and combating infectious diseases: antibodies as drug. In addition, they focus the review on an administration technique that is not widely extended in the clinic practice: electroporation. The immunization with antibodies has an immediate result of protection. As opposed to traditional vaccines that require the development of an immune response on the part of the host, taking more time to achieve immunization against diseases. This advantage is obscured by some disadvantages as administration route, short life of mAbs once injected, high cost production or cold chain storage requirement. To overcome these limitations, research has been aimed at trying to express the antibodies within the host, using nucleic acid-based technologies to express the different chains of Abs.

The review is well written, dealing with a relevant and current topic and suitable for this journal. Little aspects can be improved by the suggestions detailed below:

1.- The authors could reconsider the sentence “To date, mAbs against SARS-25 CoV-2 are the only treatments clinically proven to prevent hospitalization and death for 26 individuals infected by or exposed to the virus” (lines 25-26), due to the new antiviral data.

2.- Please, references should be included to support the sentences the occurrence of deleterious adverse 129 events (including death) during recent clinical trials evaluating AAV-mediated gene re-130 placement for rare diseases” in lines 129 and 130; in paragraph 4 (lines 142-147); and paragraph between lines 236 and 241.

3.- Could the authors comment on the difficulty of implementing electroporation among the population as a method of therapy or vaccine application? Do they see it feasible? How would this route of administration be applied to people? (headland 10)

4.- A summary table highlighting the experimental as well as clinical trials carried out by electroporation of pDNA, both to express antibodies and/or antigens, with their corresponding publications could be included.

5.- Are there trials where pDNA-NAb is applied by EP post-challenge? Would it be effective as a treatment once the disease started? Could the authors comment this aspects in headland 8.

Author Response

Review of the manuscript entitled “In vivo electroporation of plasmid DNA: a promising strategy 1 for rapid, inexpensive, and flexible delivery of anti-viral mono-2 clonal antibodies” (#1425077).

          This review overviews the currently developed antibody delivery strategies. This article highlights a little-used technique of preventing and combating infectious diseases: antibodies as drug. In addition, they focus the review on an administration technique that is not widely extended in the clinic practice: electroporation. The immunization with antibodies has an immediate result of protection. As opposed to traditional vaccines that require the development of an immune response on the part of the host, taking more time to achieve immunization against diseases. This advantage is obscured by some disadvantages as administration route, short life of mAbs once injected, high cost production or cold chain storage requirement. To overcome these limitations, research has been aimed at trying to express the antibodies within the host, using nucleic acid-based technologies to express the different chains of Abs.

The review is well written, dealing with a relevant and current topic and suitable for this journal. Little aspects can be improved by the suggestions detailed below:

1.- The authors could reconsider the sentence “To date, mAbs against SARS-25 CoV-2 are the only treatments clinically proven to prevent hospitalization and death for 26 individuals infected by or exposed to the virus” (lines 25-26), due to the new antiviral data.

 We are not exactly sure which new antiviral data the reviewer is referring to, but as of the writing of this manuscript neutralizing mAbs are the only approved or authorized treatment with demonstrated efficacy in preventing hospitalization.  Remdesivir is the only other approved therapeutic, but it is indicated for use only in hospitalized patients and has been shown to reduce the time to recovery.  Remdesivir is not approved for preventing hospitalization.

2.- Please, references should be included to support the sentences “the occurrence of deleterious adverse 129 events (including death) during recent clinical trials evaluating AAV-mediated gene re-130 placement for rare diseases” in lines 129 and 130; in paragraph 4 (lines 142-147); and paragraph between lines 236 and 241.

Regarding lines 129-130, agreed.  To our knowledge the results of this study have not been published yet (it is still ongoing) and there are no results posted on ClinicaTrials.gov, but we have cited the trial in the text.

Regarding lines 142-147, we thank review for the comment and agree that this warrants references when set apart as an independent section, which was not the intent.  This is an introductory statement for subsequent sections and there are references to support it in the sections that follow.  We have reorganized the section headings in order to make this more clear.

Regarding lines 326-241, agreed and we have added the appropriate references.

3.- Could the authors comment on the difficulty of implementing electroporation among the population as a method of therapy or vaccine application? Do they see it feasible? How would this route of administration be applied to people? (headland 10)

We thank the reviewer for this comment, although we cannot say much in response as this is largely unknown.  As with any new technology, there will undoubtedly be certain barriers to its initial implementation, though because of the potential benefits outlined in this manuscript we do believe it is worth pursuing and ultimately feasible.

In terms of how it would be applied to people, we think that is also largely yet to be determined. During clinical trial phases, it will be administered in a clinical setting such as a hospital or doctor’s office, but the administration conditions after commercialization could potentially be expanded to additional settings, much like those where vaccinations might be given. 

4.- A summary table highlighting the experimental as well as clinical trials carried out by electroporation of pDNA, both to express antibodies and/or antigens, with their corresponding publications could be included.

While we agree that such a table would be interesting, since the focus of this review is on the delivery of antibodies, and there is currently only 1 clinical trial evaluating the delivery of DNA-encoded antibodies by DNA/EP, we believe such a table would be a bit misplaced in this manuscript.

5.- Are there trials where pDNA-NAb is applied by EP post-challenge? Would it be effective as a treatment once the disease started? Could the authors comment this aspects in headland 8.

This is an excellent point and we thank the reviewer for these questions.  There are no clinical trials where EP is used to administer pDNA-NAbs post challenge.  In terms of the potential for that type of use for this technology, it would depend on the kinetics of the disease course.  For short-term, acute viral infections such as SARS-CoV-2, it is unlikely to be beneficial as a therapeutic because the peak serum antibody levels take 2-3 weeks to build up, which is well past the acute viral phase of the disease.  If, however, the antibody was potent enough that peak levels weren’t needed for efficacy, EP might be a quite viable treatment strategy.  For chronic viral infections, such as HIV-1, EP could reasonably be used for treatment post-infection.  Text has been added to address this point in the recommended section.

Reviewer 2 Report

The authors provide a comprehensive overview of iIn vivo electroporation of plasmid DNA for delivery of anti-viral mAbs, which is of interest for a broad readership. Though the authors are very enthusiastic about the prospects, they discuss major efficacy/safety concerns and limitations regarding various delivery systems as well.

Prior to final acceptance, I would like to recommend some minor revisions:

  1. Concerning the safety of introducing plasmid DNA into host cells, the author cite a study (ref. 54) that the injected pDNA remains essentially at the site of injection for a mean period of two months, followed by eventual clearance in the majority of animals. Is there clear proof that no insertion of plasmid DNA occurs into the host genome under all circumstances?
  2. Antibody-dependent enhancement (ADE) as a major efficacy/safety concern needs to be briefly discussed for Abs against the Covid spike protein as well, thus limiting the immune response, the more so as the spike protein has been shown to damage endothelial cells (e.g. Lei et al. Endothelial cell damage is the central part of COVID-19 and a mouse model induced by injection of the S1 subunit of the spike protein. Res. 2021 128(9):1323-1326. doi: 10.1161/CIRCRESAHA.121.318902.)
  3. „The recent technical and medical success of mRNA vaccines against COVID-19..“ and other similar statements regarding the efficacy/safety of mRNA vaccines against COVID-19 by the authors is not appropriate in light of recent experiences as exemplified below:

    3a. The Office for National Statistics (ONS) in the United Kingdom has released a new dataset (https://www.ons.gov.uk/peoplepopulationandcommunity/birthsdeathsandmarriages/deaths/articles/deathsinvolvingcovid19byvaccinationstatusengland/deathsoccurringbetween2januaryand2july2021) showing that 81 percent of all people who died in the month of September after testing “positive” for Covid-19 was “fully vaccinated” in accordance with government guidelines. In the U.K. alone during the month of September, some 30,305 people died within 21 days of getting vaccinated against COVID-19.

    3b. Two different studies in press (published in NEJM!) from Israel and Qatar have demonstrated that antibody levels wane within a few months after receiving Pfizer’s COVID-19 vaccine and that infection rates are much higher among vaccinated individuals. (Levin et al. Waning Immune Humoral Response to BNT162b2 Covid-19 Vaccine over 6 Months. N. Engl. J. Med. 2021 Oct 6. doi: 10.1056/NEJMoa2114583.  Chemaitelly et al. Waning of BNT162b2 Vaccine Protection against SARS-CoV-2 Infection in Qatar. N. Engl. J. Med. 2021 Oct 6. doi: 10.1056/NEJMoa2114114.)

    3c. Similarly, a recent paper (Shitrit et al. Nosocomial outbreak caused by the SARS-CoV-2 Delta variant in a highly vaccinated population, Israel, July 2021. Euro Surveill. 2021;26(39):pii=2100822. https://doi.org/10.2807/1560-917.ES.2021.26.39.2100822) reports a COVID outbreak in an Israeli hospital, despite a 96% vaccination rate and use of protective equipment. The calculated rate of infection among all exposed patients and staff was 10.6% (16/151) for staff and 23.7% (23/97) for patients. Of the infected, 23 were patients and 19 were staff members. While all the staff recovered quickly, eight vaccinated patients became severely ill, six became critically ill and five of the critically ill died. The two unvaccinated patients tracked had mild COVID cases.

Author Response

The authors provide a comprehensive overview of iIn vivo electroporation of plasmid DNA for delivery of anti-viral mAbs, which is of interest for a broad readership. Though the authors are very enthusiastic about the prospects, they discuss major efficacy/safety concerns and limitations regarding various delivery systems as well.

We thank the reviewer for this comment.  We have tried to be comprehensive in addressing the topic of this review.

Prior to final acceptance, I would like to recommend some minor revisions:

  1. Concerning the safety of introducing plasmid DNA into host cells, the author cite a study (ref. 54) that the injected pDNA remains essentially at the site of injection for a mean period of two months, followed by eventual clearance in the majority of animals. Is there clear proof that no insertion of plasmid DNA occurs into the host genome under all circumstances?

Genomic insertion of plasmid DNA has certainly not been tested in “all circumstances” so no, there is no proof of such.  We have included a reference in which data are presented the support the conclusion that there is currently no evidence of genomic insertion of plasmid DNA, at least in the context of DNA vaccines.

  1. Antibody-dependent enhancement (ADE) as a major efficacy/safety concern needs to be briefly discussed for Abs against the Covid spike protein as well, thus limiting the immune response, the more so as the spike protein has been shown to damage endothelial cells (e.g. Lei et al. Endothelial cell damage is the central part of COVID-19 and a mouse model induced by injection of the S1 subunit of the spike protein. Res. 2021 128(9):1323-1326. doi: 10.1161/CIRCRESAHA.121.318902.)

ADE, while certainly a concern for some viral diseases, is a more relevant topic for a review on antibody selection, rather than the technologies by which antibodies are delivered.  We do not believe it is necessary to weigh in on ADE concerns in this manuscript.

  1. „The recent technical and medical success of mRNA vaccines against COVID-19..“ and other similar statements regarding the efficacy/safety of mRNA vaccines against COVID-19 by the authors is not appropriate in light of recent experiences as exemplified below:

3a. The Office for National Statistics (ONS) in the United Kingdom has released a new dataset (https://www.ons.gov.uk/peoplepopulationandcommunity/birthsdeathsandmarriages/deaths/articles/deathsinvolvingcovid19byvaccinationstatusengland/deathsoccurringbetween2januaryand2july2021) showing that 81 percent of all people who died in the month of September after testing “positive” for Covid-19 was “fully vaccinated” in accordance with government guidelines. In the U.K. alone during the month of September, some 30,305 people died within 21 days of getting vaccinated against COVID-19. .

3b. Two different studies in press (published in NEJM!) from Israel and Qatar have demonstrated that antibody levels wane within a few months after receiving Pfizer’s COVID-19 vaccine and that infection rates are much higher among vaccinated individuals. (Levin et al. Waning Immune Humoral Response to BNT162b2 Covid-19 Vaccine over 6 Months. N. Engl. J. Med. 2021 Oct 6. doi: 10.1056/NEJMoa2114583.  Chemaitelly et al. Waning of BNT162b2 Vaccine Protection against SARS-CoV-2 Infection in Qatar. N. Engl. J. Med. 2021 Oct 6. doi: 10.1056/NEJMoa2114114.)

3c. Similarly, a recent paper (Shitrit et al. Nosocomial outbreak caused by the SARS-CoV-2 Delta variant in a highly vaccinated population, Israel, July 2021. Euro Surveill. 2021;26(39):pii=2100822. https://doi.org/10.2807/1560-917.ES.2021.26.39.2100822) reports a COVID outbreak in an Israeli hospital, despite a 96% vaccination rate and use of protective equipment. The calculated rate of infection among all exposed patients and staff was 10.6% (16/151) for staff and 23.7% (23/97) for patients. Of the infected, 23 were patients and 19 were staff members. While all the staff recovered quickly, eight vaccinated patients became severely ill, six became critically ill and five of the critically ill died. The two unvaccinated patients tracked had mild COVID cases.

Similar to the previous point, we do not believe that significant commentary on the safety and efficacy of mRNA vaccines against COVID-19 is warranted in a review article on nucleic acid-based antibody delivery technology.  However, a general statement regarding the successes of these vaccines reflects the overwhelming consensus of the scientific and medical communities and is appropriate in this review.